# Matching in Multi-arm Bandit with Collision

**Yirui Zhang**[1], **Siwei Wang**[2], **Zhixuan Fang**[1,3*]

[1] IIIS, Tsinghua University  [2] Microsoft Research  [3] Shanghai Qi Zhi Institute

zhangyr22@mails.tsinghua.edu.cn
siweiwang@microsoft.com
zfang@mail.tsinghua.edu.cn

## Abstract

In this paper, we consider the matching of multi-agent multi-armed bandit problem, i.e., while agents prefer arms with higher expected reward, arms also have preferences on agents. In such case, agents pulling the same arm may encounter collisions, which leads to a reward of zero. For this problem, we design a specific communication protocol which uses deliberate collision to transmit information among agents, and propose a layer-based algorithm that helps establish optimal stable matching between agents and arms. With this subtle communication protocol, our algorithm achieves a state-of-the-art $O(\log T)$ regret in the decentralized matching market, and outperforms existing baselines in experimental results.

## 1   Introduction

Decentralized matching between two sides with preferences (e.g., supply and demand) is the core process in many online marketplaces, such as passengers and riders on ridesharing platforms (e.g., Uber), or customers and free-lancers in online labor markets (e.g., TaskRabbit), etc. Since in practical scenarios, participants in the online market obtain information mainly through their own experiences, a crucial question arises on *how to achieve an optimal stable matching among these competing agents, through learning from the iterative interactions with the other side, and, at what cost?*

Learning the uncertainty in two-sided markets has been studied from various angles, e.g., the contextual bandit scenario ([8, 7]), the economic perspective [1], and the stability and fairness in the centralized matching market [5]. Following a strand of recent literature (e.g., [9, 2]), we model the problem as a multi-agent multi-arm bandit problem. Specifically, in the decentralized matching market with multi-armed bandits, we consider a set $\mathcal{M}$ of $M$ agents, and a set $\mathcal{A}$ of $K$ arms, with $M \leq K$. Every agent knows $M$, $K$, and the time horizon $T$. Each arm has its fixed preference towards agents, which is unknown to all the agents. The notation $i \succ_k j$ means that arm $k$ prefers agent $i$ rather than agent $j$.

At each time $t$, every agent $j$ pulls an arm $I_j(t) \in \mathcal{A}$. When multiple agents pull the same arm $k$ at time $t$, a collision occurs on this arm. Only the agent that the arm prefers most, e.g., $j$, will win the competition. The winning agent does not encounter collision, and obtains a stochastic reward $R_j(t)$ from the arm, which is sampled from a unknown fixed distribution $F_{jk}$ with mean $u_{jk}$. In the meantime, the other agents $i$ that pull arm $k$ will encounter collision and receive zero reward, i.e., $R_i(t) = 0$. Same with the previous works, we assume that the minimal gap of utility between two arms among all agents $\Delta \triangleq \min_{m \in \mathcal{M}} \min_{k,n \in \mathcal{A}, k \neq n} |u_{mk} - u_{mn}|$ is positive. We let $C_j(t)$ represent the collision indicator for agent $j$ at time $t$, i.e., $C_j(t) = 1$ denotes that agent $j$ encounters collision at time step $t$ and $C_j(t) = 0$ otherwise. In our model, each agent $j$ is informed of her own collision signal $C_j(t)$ and reward $R_j(t)$ after her pull. At each time step $t$, every agent $j$ pulls an arm $I_j(t)$ only according to her observed history $\{(I_j(\tau), R_j(\tau), C_j(\tau))\}_{\tau=1}^{t-1}$.

---

*Corresponding author: Zhixuan Fang (zfang@mail.tsinghua.edu.cn).

36th Conference on Neural Information Processing Systems (NeurIPS 2022).

Table 1. Comparison between our work and prior results.

| Algorithms | Model | Regret |
|---|---|---|
| UCB-D3 | globally ranked agent | $O(M(M-1)K\log T/\Delta^2)$ |
| UCB-D4 | uniqueness consistency | $O(M(M-1)K\log T/\Delta^2)$ |
| CA-UCB | partially decentralized | $O(\exp(M^4)M\log^2 T/\Delta^2)$ (pessimal regret) |
| Phased-ETC | general model | $O(MK\log^{1+\epsilon} T + \exp(1/\Delta^2))$ |
| ML-ETC (our result) | general model | $O(MK\log T/\Delta^2)$ |

Now we introduce the optimal stable matching to give a suitable notion of regret ([6]). While arms have preferences towards agents, an agent $j$ always prefers the arm with higher utility, i.e., the arm with higher $u_{jk}$. In a stable matching, there does not exists any "agent, arm" pair such that each one prefer each other than the current matched partner.

If there exists at least one stable matching where agent $j$ is matched with arm $k$, we call arm $k$ is an attainable arm to agent $j$. We denote the set of attainable arms of agent $j$ by $\mathcal{A}_{stable}(j)$. The optimal stable arm for agent $j$ is $a_j^* = \operatorname{argmax}_{k \in \mathcal{A}_{stable}(j)} u_{jk}$, i.e., the arm with highest mean to agent $j$ in $\mathcal{A}_{stable}(j)$. As shown in previous study (e.g.,[6]), there exists a unique stable matching where every agent matches with her optimal stable arm. The optimal regret is defined as the expected reward gap between the ideal reward according to the optimal stable matching and the achieved reward, i.e.,

$$Reg(T) = T\sum_{j=1}^{M} u_{ja_j^*} - \mathbb{E}\left[\sum_{t=1}^{T}\sum_{j=1}^{M}(1 - C_j(t))u_{jI_j(t)}\right].$$

## 1.1 Our Main Contribution

Following the explore-then-commit (ETC) framework [11, 9, 12], we propose an algorithm called ML-ETC. Specifically, all the agents will first explore arms and then commit to their optimal stable arms in terms of empirical means.

The key point in ML-ETC is that we allow agents to communicate through deliberately designed collisions, so that they can switch from exploration to exploitation timely. Previous works in multi-agent multi-arm bandit (e.g., [4, 15, 11, 14, 3]) have discussed the use of deliberate collisions among agents for communication, but they fail in the case of matching market with multi-armed bandit. This is because when arms have preferences on agents, it may be impossible for every agent to communicate with each other. For example, consider an extreme case where agent $j$ is ranked first on all arms. In this case, agent $j$ will never encounter collision, let alone receiving information from others through deliberate collisions. To solve this problem, we show that under such circumstance, the agent who cannot transmit information to agent $j$ will never be able to influence the stable matching of agent $j$, which means that there is nearly no difference whether they exist from agent $j$'s point of view. Therefore, an agent could end the exploration once all the agents who can communicate with her have high confidence of their preference about arms. After that, we can apply GS ([6]) to make sure that each agent obtains her optimal stable arm within limited time steps and then commits to that optimal stable arm.

Based on these properties, we show that our subtle multi-layered algorithm ML-ETC achieves a state-of-the-art regret upper bound of $O(\log T)$ in the decentralized setting, which is better than the existing results, including the $O(\log^{1+\epsilon} T)$ regret bound for Phased-ETC [2] and the $O(\log^2 T)$ regret bound for CA-UCB in [10]. Please see Table 1 for the summary of comparison with previous results, and Section 1.2 for detailed discussions.

## 1.2 Related Work

The study of matching in multi-armed bandit is initiated by the recent work [9]. In this work, the authors introduce the model and design both a centralized ETC policy and a decentralized ETC policy. The centralized ETC has good performance but requires a central organizer to proceed matching, which may not exist in many real-world applications. The decentralized ETC, on the other hand, may cause very high regret when the minimal gap $\Delta$ is unknown to the agents, which is also a common case in reality. Thereafter, there are two major strands of literature on this problem.

A strand of literature studies the special case where the arms' preferences satisfy some special conditions. For example, the work of [13] assumes that the agents are globally ranked, which means all arms have the same preference toward agents, and proposes the UCB-D3 algorithm. Also, [2] designs the UCB-D4 algorithm based on the assumption of uniqueness consistency on agents and arms. The assumption of uniqueness consistency requires that there exists a unique stable matching for all agents and arms, and the matching remains to be the unique stable matching when arbitrary matched pairs leave the system.

The other strand of literature focuses on the general case of unrestricted arm preferences. In [10], the authors design the CA-UCB algorithm, based on extra information that all agents can observe the winning agent for each arm at the end of each round. CA-UCB achieves an $O(\log^2 T)$ pessimal regret (a less strict notion of regret). Within this strand of work, the work [2] is the most related to our work, which also analyzes the optimal regret in the general market. Beyond the aforementioned UCB-D4, the authors also design a phased based algorithm Phased-ETC (for general arm preference) and achieves the regret bound of $O(\log^{1+\epsilon} T)$, with $\epsilon > 0$. However, in their algorithm, agents have no communication at all. Therefore, agents keeps exploring all the arms and have no idea of others' estimations of the arms. In addition to the $O(\log^{1+\epsilon} T)$ factor, Phased-ETC also has a huge regret term (an exponential term of $\exp(\frac{1}{\Delta^2})$) because of the possibility of the exploration failure, and this will be explained in detail in our experimental section. Compared to [2], we design a subtle communication protocol to allow agents to end the exploration timely while they have high confidence of the arms' order according to the empirical means. This allows us to obtain a state-of-the-art $O(\log T)$ regret upper bound without any potential exponential terms.

## 2 The ML-ETC Algorithm

In this section, we introduce our Multi-Layer Explore Then Commit (ML-ETC) algorithm, which is described in detail in Algorithm 1 and achieves $O(\log T)$ regret.

### 2.1 Brief Introduction

The algorithm can be briefly divided into 3 phases: the *initialization* phase, the *multi-layered exploration and communication* phase, and the *exploitation* phase.

In the initialization phase, all agents will be assigned an index and receive some information of the arm preferences. After the initialization phase, agents form an underlying hierarchical structure with respect to their communication capability through collision. Then, the algorithm proceeds to the multi-layered exploration and communication phase to match agents of each layer with arms. At each layer, the algorithm keeps iterating between exploration and communication. During exploration, every agent will explore the arms orthogonally (according to her index in order to avoid collision) and will estimate the expected rewards of all the remaining arms. During communication, every agent checks whether her estimations are accurate enough, and transmit this information to all the agents that can hear from her. Only when some of the agents are able to successfully find their empirical optimal stable arms, will they leave the multi-layered exploration and communication phase, and start their exploitation phase. The remaining agents will enter the next layer, until all the agents find out their empirical optimal stable arms. In the exploitation phase, every agent continuously pulls her empirical optimal stable arm.

### 2.2 Communication Protocol

In this subsection, we will introduce the core of ML-ETC, the communication protocol used in both the initialization phase and the multi-layered exploration and communication phase.

Suppose agent $i$ (i.e., the transmitter) wants to convey a binary sequence $s$ to agent $j$ (i.e., the receiver). Let $|s|$ denote the length of the binary sequence, i.e., $s \in \{0,1\}^{|s|}$. We design a procedure $com(i, j, k, s)$ to allow such communication, where agent $i$ and agent $j$ will create deliberate collisions on arm $k$ to deliver the bits. The communication procedure will last for $|s|$ time steps, during which other agents are serving as bystanders. We specify the actions of all agents in $com(i, j, k, s)$ as follows.

- Transmitter $i$: During the procedure $com(i, j, k, s)$, at the $\ell$-th time step, if $s(\ell)$ (the $\ell$-th digit in $s$) is 1, then the transmitter $i$ will pull arm $k$. Otherwise, she pulls an arbitrary arm $k' \neq k$ at the time step.
- Receiver $j$: During $com(i, j, k, s)$, the receiver $j$ deliberately pulls arm $k$ for all $|s|$ time steps to receive information, and the resulting collision indicators $C_j(t)$ during these time steps are the digits of the binary sequence $s$.
- Bystanders: During $com(i, j, k, s)$, all agent $m \neq i, j$ are bystanders. Each bystander chooses an arbitrary arm $k' \neq k$ to pull for each time step within all $|s|$ time steps of the procedure, to avoid interfering the bit transmission between agent $i$ and $j$ on arm $k$.

In this paper, agents may transmit two types of information: the index of an agent, a binary signal of whether the exploration is success or not. It is easy to check that the communication of these two types of information are with length $\lceil \log M \rceil$ and 1.

However, $com(i, j, k, s)$ does not always succeed. Only if $i \succ_k j$, i.e., when agent $i$ wins against $j$ during the competition on arm $k$, the collision indicators $C_j(t)$'s form the sequence $s$. Otherwise, $C_j(t)$ always equals to 0 and therefore $com(i, j, k, s)$ is not effective. Based on this observation, we define the communication graph as follows.

**Definition 1.** *A communication graph $G(\mathcal{M}, \mathcal{A})$, where $\mathcal{M}$ is the set of agents and $\mathcal{A}$ is the set of arms, is a directed graph with each agent in $\mathcal{M}$ as one vertex. There is a directed edge from vertex $i$ to vertex $j$ in $G$ if and only if there exists at least one arm $k \in \mathcal{A}$ such that $i \succ_k j$.*

Based on Definition 1, if there exists at least one edge from $i$ to $j$ in graph $G(\mathcal{M}, \mathcal{A})$, it means that agent $i$ can transmit information to agent $j$ effectively (through the proper arm). The communication graph captures the communication availability among agents. We will further categorize agents with similar communication "capability" as follow.

**Definition 2.** *An outer closed circle is a non-empty subset of vertexes in a directed graph $G$, i.e., $M \subseteq V(G)$, which satisfies that: i) $M$ is connected; ii) there is no entry edge from vertices in $V(G) \setminus M$ to any vertex in $M$.*

From Definition 2, we can easily see that the agents in an outer closed circle of the communication graph $G(\mathcal{M}, \mathcal{A})$ can exchange (transmit and receive) information among each other. However, agents in the outer closed circle can only transmit information to agents outside, but are unable to receive information from them. The following proposition further shows that there always exists exactly one outer closed circle in a communication graph $G(\mathcal{M}, \mathcal{A})$.

**Proposition 1.** *If there exists at least one edge between any two vertices in directed graph $G$, there exists a unique outer closed circle in $G$.*

Due to space limit, the details of the proof of Proposition 1 is deferred to Appendix in the supplementary material.

**Remark: Multi-layer separation of agents.** The result of Proposition 1 is critical. With this result, we can uniquely determine a group of agents with bidirectional communication capability within the group, but unidirectional transmission to the outside. Thus, by iteratively extracting outer closed circle from the agent set $\mathcal{M}$, we are naturally constructing a hierarchical structure within agents. In such structure, agents can transmit information only to agents of the same level and below, while receiving information from agents of the same level and higher (see Section 2.4 for detailed discussions).

### 2.3   Initialization Phase

The initialization phase (lines 1-4 in Algorithm 1) consists of two procedures: $index\_assignment$ and $information\_access$. Through procedure $index\_assignment$, each agent is assigned an index, and obtains some information about each arm's preferences. Then, the procedure $information\_access$ allows agents to share their obtained preference rankings on all arms.

The procedure $index\_assignment$ (see details in Appendix) consists of $K$ rounds with $M$ time steps each, lasting a total $MK$ time steps. In round 1, every agent starts with pulling arm 1. If an agent does not encounter collision at time step $t = j_1$ ($j_1 \in [1, M]$), the agent is assigned with index $j_1$ and starts to pull arm 2. This means that the index of an agent is her ranking on arm 1. Similarly, in round $k = 2, 3, ..., K$, if an agent does not encounter collision on arm $k$ at time step

---
**Algorithm 1** ML-ETC Algorithm
---
**Require:** $K, M, T, p = 1$
 1: ◁ Initialization phase:
 2: $(j, s_j) \leftarrow index\_assignment(K, M)$ #assign index $j$ to "this agent"
 3: $(s_j, \mathcal{M}_p) \leftarrow information\_access(K, M, j, s_j)$
 4: #$s_j$ is the arm preference known to agent $j$, $\mathcal{M}_p$ is the set of layer leaders in layer $p$
 5: ◁ Multi-layered exploration and communication phase:
 6: $E(j) \leftarrow 0$ # $E(j)$ denotes whether agent $j$ enter the exploitation phase
 7: **while** $E(j) == 0$ **do**
 8:    ◁ Communication:
 9:    **if** $j \in \mathcal{M}_p$ **then** #Agent $j$ is a layer leader
10:       $whether\_success \leftarrow whether\_succ(\hat{\boldsymbol{u}}(j), N(j), \mathcal{A}_p)$
11:       #whether $j$ achieves an accurate estimation for all remaining arms $\mathcal{A}_p$
12:       #$\mathcal{A}_p$ is the set of remaining arms in layer $p$, $\mathcal{R}_p$ is the set of remaining agents in layer $p$
13:       $(opt\_arm, E(j)) \leftarrow layer\_leader(j, s_j, \mathcal{A}_p, \mathcal{R}_p, \mathcal{M}_p, whether\_success, M)$
14:       #if all the layer leaders succeed, then $E(j) \leftarrow 1$ and $j$ will enter the exploitation phase
15:       #$opt\_arm$ is the empirical optimal stable arm of agent $j$
16:    **end if**
17:    **if** $j \notin \mathcal{M}_p$ **then** #Agent $j$ is a layer follower
18:       $(p, \mathcal{A}_p, \mathcal{M}_p, \mathcal{R}_p) \leftarrow layer\_follower(j, s_j, \mathcal{A}_p, \mathcal{R}_p, \mathcal{M}_p, M, p)$
19:       #if all the layer leaders succeed, then $p \leftarrow p + 1$ and agent $j$ will enter the next layer
20:    **end if**
21:    ◁ Exploration:
22:    Let $r$ be the order of agent $j$ in remaining agent set $\mathcal{R}_p$, $i \leftarrow r$
23:    **for** $|\mathcal{A}_p|\lceil \log T \rceil$ time steps **do**
24:       $\pi \leftarrow \mathcal{A}_p[i]$
25:       $i \leftarrow i + 1 \mod |\mathcal{A}_p|$
26:       Pull arm $\pi$, update the empirical means $\hat{\boldsymbol{u}}(j) = \{\hat{u}_{j1}(t), \hat{u}_{j2}(t), ...\}$ and matching
27:       times $\boldsymbol{N}(j) = \{N_{j1}(t), N_{j2}(t), ...\}$
28:    **end for**
29: **end while**
30: ◁ Exploitation phase:
31: Pull $opt\_arm$ until $T$
---

$t = (k - 1)M + j_k$, she knows that her ranking on arm $k$ is $j_k$, and starts to keep pulling arm $k + 1$ mod $K$, so that the other agents can continue to confirm their rankings on arm $k$. In this way, after $MK$ time steps, every agent knows her ranking on all the arms.

Then in the procedure of $information\_access$ (given in Appendix), there will be $M$ rounds of information exchange. In each round, we list all the "agent, agent, arm" tuples, i.e., $\{(i, j, k)\}, \forall i, j \in \mathcal{M}, k \in \mathcal{A}$, and conduct communication procedure $com(i, j, k, s_i)$ on all these tuples one by one according to any publicly designated order, no matter whether the communication is effective or not. Here $s_i$ represents agent $i$'s updated knowledge about all rankings, e.g., $s_i \in \{0, 1, ..., M\}^{K \times M}$, where the matrix element $s_i[n][m]$ is the index of the agent that has the $m$-th ranking on arm $n$. If the index is unknown to agent $i$, $s_i[n][m]$ is 0. Each matrix element $s_i[n][m]$ is encoded in a binary sequence with length $\lceil \log M \rceil$ for transmission.

After the initialization phase, every agent $i$ will receive an index and get part of the information about agents' rankings. Specifically, agent $i$ receives information of the rankings of all the agents $j$ such that there exists a path from $j$ to $i$ in $G(\mathcal{M}, \mathcal{A})$, i.e., agents of the same level as $i$ or higher.

## 2.4 Multi-layered exploration and communication phase

The multi-layered exploration and communication phase (lines 5-29 in Algorithm 1) is driven by the layer categorization of agents. Recall Proposition 1, we can first identify a unique subset of agents from $\mathcal{M}$, called an outer closed circle, in which agents can communicate with each other. At this stage, we call the system is at "layer 1". Agents in the outer closed circle are called layer leaders of layer 1, the set of which is denoted as $\mathcal{M}_1$. Our algorithm will finish this stage by assigning each

agent in $\mathcal{M}_1$ her empirical optimal stable matching arm. Thus, agents in $\mathcal{M}_1$ will leave this stage and directly jump to the exploitation phase. Those remaining agents and arms will enter the stage of layer 2 and start the identification of a new outer closed circle. The above process will keep repeating until all agents enter the exploitation phase.

In general, for each layer $p = 1, 2, 3, ...$ the system is in, we let $\mathcal{R}_p$ denote the set of all remaining agents, and $\mathcal{A}_p$ denote the set of all remaining arms (we have $\mathcal{R}_1 = \mathcal{M}$ and $\mathcal{A}_1 = \mathcal{A}$). Then according to Proposition 1, there exist a group of agents $\mathcal{M}_p$, which form the outer closed circle of the communication graph $G(\mathcal{R}_p, \mathcal{A}_p)$. We call the agents in $\mathcal{M}_p$ layer leaders in this layer, while the remaining agents in $\mathcal{R}_p \setminus \mathcal{M}_p$ are called layer followers. Based on the properties of the outer closed circle, all the layer leaders can i) exchange information among each other, and ii) transmit information to all layer followers but not be able to receive information from them (via the remaining arms in $\mathcal{A}_p$). Then, the goal in this layer is to let all the layer leaders find out their empirical optimal stable arm. The reason that we only care about layer leaders is because the communication from layer followers to layer leaders is not effective, and layer followers can not influence layer leaders (including their pull decisions, results and optimal stable arms).

To achieve this goal, at any layer, all remaining agents will conduct communication and exploration. They will first communicate with each other to check whether all the layer leaders have obtained an accurate estimation on the remaining arms. If the layer leaders have obtained accurate estimation on all remaining arms, then they will enter the exploitation phase, otherwise, they will conduct exploration for $|\mathcal{A}_p|\lceil \log T \rceil$ time steps. Specifically, for a layer leader $i$, if for any two remaining arms $m, n$, either the lower confidence bound of $u_{jn}$ is larger than the upper confidence bound of $u_{jm}$ or the lower confidence bound of $u_{jm}$ is larger than the upper confidence bound of $u_{jn}$ (i.e., with high probability, the order of empirical means is the same as the order of real means), then she can confidently differentiate these two arms. Thus, the agent has successfully obtained an estimation (or, a sort) on all remaining arms that is accurate enough for her to enter the exploitation phase. In the communication, we let layer leaders reach a consensus on a global signal of whether all layer leaders have achieved a successful estimation on every remaining arm, and transmit this signal to all layer followers, while layer followers do not transmit anything. Since all the layer leaders can hear from each other, after the communication, they know the updated status of all layer leaders. If all layer leaders successfully sort the arms with confidence, they start to find out their empirical optimal stable-matching arms (by conducting the $GS\_and\_arm\_information$ procedure based on the Gale-Shapley algorithm), and then jump to the exploitation phase. Otherwise, all agents will start a round of exploration, so that the agents with unsuccessful signal of exploration have the chance to achieve an accurate estimation successfully before the next communication. When all the layer leaders jump to the exploitation phase, all the remaining agents (the layer followers in this layer) start to enter the next layer and learn the remaining arm set.

Note that at the beginning of layer $p$, a remaining agent $i$ is able to identify all layer leaders and layer followers by her own, based on the knowledge of $\mathcal{R}_p$, $\mathcal{A}_p$ and her available information about the rankings from the initialization phase (See Appendix for detailed proof). Therefore, the above procedure can be done in a distributed manner. For simpler presentation, in the rest of this subsection, we will first specify the exploration process, and then the communication process.

### 2.4.1 Exploration

Within one round of exploration at layer $p$ that lasts for $|\mathcal{A}_p|\lceil \log T \rceil$ time steps, every agent in $\mathcal{R}_p$ will explore every remaining arm in $\mathcal{A}_p$ orthogonally according to their index to avoid collision. Specifically, at the $\tau$-th time step in this round of exploration, the $j$-th agent in $\mathcal{R}_p$ will pull the $(j + \tau \mod |\mathcal{A}_p|)$-th arm in $\mathcal{A}_p$. Since $|\mathcal{A}_p| \geq |\mathcal{R}_p|$, every agent will pull different arms through the exploration and avoid collision with each other.

The empirical mean of arm $k$ estimated by agent $j$ during exploration after time step $t$ is denoted by $\hat{u}_{jk}(t)$, the number of time steps that arm $m$ is pulled by agent $j$ in exploration at the end of time step $t$ is denoted by $N_{jm}(t)$. We say agent $j$ achieves an accurate estimation for the remaining arms in $\mathcal{A}_p$ successfully at time step $t$ (i.e., she will transmit a success signal of exploration) only if for any pair of arms $(m, n)$ in $\mathcal{A}_p$, either $\hat{u}_{jm}(t) - \sqrt{\frac{2 \log T}{N_{jm}(t)}} > \hat{u}_{jn}(t) + \sqrt{\frac{2 \log T}{N_{jn}(t)}}$ or $\hat{u}_{jn}(t) - \sqrt{\frac{2 \log T}{N_{jn}(t)}} > \hat{u}_{jm}(t) + \sqrt{\frac{2 \log T}{N_{jm}(t)}}$ is satisfied.

### 2.4.2 Communication

---

**Algorithm 2** layer_leader

---

**Require:** $j, s_j, \mathcal{A}_p, \mathcal{R}_p, \mathcal{M}_p, whether\_success, M$
**Ensure:** $opt\_arm, E(j)$
1: $opt\_arm \leftarrow 1$, $E(j) \leftarrow 0$, $all\_leader\_success \leftarrow whether\_success$
2: **for** $m = 1, 2, ..., |\mathcal{M}_p|, i = 1, 2, ..., |\mathcal{A}_p|, i_1 = 1, 2, ..., M, i_2 = i_1 + 1, ..., M$ **do**
3:      **if** $s_j[\mathcal{A}_p[i]][i_1] == j$ **then**
4:          If $all\_leader\_success == 0$, agent $j$ pulls arm $i$; otherwise pulls an arbitrary arm $i' \neq i$.
5:          # agent $j$ is a transmitter
6:      **else if** $s_j[\mathcal{A}_p[i]][i_2] == j$ **then**
7:          Agent $j$ pulls arm $i$, and only if $C_j(t) == 1$, she sets $all\_leader\_success = 0$.
8:          # agent $j$ is a receiver
9:      **else**
10:         Agent $j$ pulls an arbitrary arm $i' \neq i$. # agent $j$ is a bystander
11:      **end if**
12: **end for**
13: **if** $all\_leader\_success == 1$ **then**
14:      $opt\_arm \leftarrow GS\_and\_arm\_information(j, \hat{\boldsymbol{u}}(j), \mathcal{A}_p)$
15:      $E(j) \leftarrow 1$
16: **end if**
17: **return** $opt\_arm, E(j)$

---

**Layer leaders.** For layer leaders at layer $p$, their communication procedure is described in Algorithm 2. From lines 1-12, they exchange their information about whether all the layer leaders achieve an accurate estimation successfully and transmit this information to the layer followers at the mean time.

If all layer leaders achieve an accurate estimation successfully, then they continue to execute $GS\_and\_arm\_information$ and enter the exploitation phase. Otherwise, they start a round of exploration according to ML-ETC. The $GS\_and\_arm\_information$ procedure in Algorithm 2 lasts for $|\mathcal{M}_p|^2 + (|\mathcal{R}_p| - |\mathcal{M}_p|)|\mathcal{A}_p|$ time steps. In the first $|\mathcal{M}_p|^2$ time steps, layer leaders follow the standard Gale–Shapley algorithm [6], where each agent's preference towards arms is in the descending order of the empirical means. Thus, each layer leader proposes to the best unrejected arm iteratively (i.e., the empirical best arm on which agent hasn't encounter a collision yet). This makes sure that all the agents obtain their empirical optimal stable arms. In the rest $(|\mathcal{R}_p| - |\mathcal{M}_p|)|\mathcal{A}_p|$ time steps, they commit to their empirical optimal stable arms so that the layer followers have enough time to receive the information about these "leaving arms" (which is used to construct the remaining arm set in the next layer).

**Layer followers.** For a layer follower in layer $p$, her actions in the communication is described in Algorithm 3. From lines 2-8, the agent either stands by to wait for her turn or to receive the information whether all layer leaders achieve an accurate estimation successfully (denoted by $all\_leader\_success$ in line 4).

If there exist some layer leaders sending the unsuccess signal of exploration, all the layer followers switch to a new round of exploration. Otherwise, a layer follower continues to execute lines 10-25. Specifically, layer followers first wait for $|\mathcal{M}_p|^2$ time steps by choosing an arbitrary arm to pull. Note that based on the property of layer followers, their actions will not influence the GS algorithm among layer leaders. Then for each arm in $\mathcal{R}_p$, layer followers explore whether the arm is available (i.e., the remaining arm set $\mathcal{A}_{p+1}$ in the next layer) based on the order of the their indices one by one. For an agent $i$, if it is not her turn to explore an arm, e.g., $k$, she will always choose the next arm in $\mathcal{A}_p$ to pull. Note that in $i$'s exploration of available arms, all the other layer followers $j \neq i$ are pulling another arm in $\mathcal{A}_p$. Therefore, those arms on which the agent $i$ gets a collision will leave at the end of this layer. Below, We illustrate the algorithm with an example.

**Example 1.** *Let $\mathcal{M} = \{A, B, C, D\}$ and $\mathcal{A} = \{1, 2, 3, 4\}$ and assume the preference below:*

$$1 : A \succ_1 B \succ_1 C \succ_1 D \qquad A : 1 \succ_A 2 \succ_A 3 \succ_A 4$$
$$2 : A \succ_2 B \succ_2 D \succ_2 C \qquad B : 1 \succ_B 3 \succ_B 2 \succ_B 4$$
$$3 : B \succ_3 A \succ_3 C \succ_3 D \qquad C : 2 \succ_C 3 \succ_C 4 \succ_C 1$$
$$4 : A \succ_4 B \succ_4 D \succ_4 C \qquad D : 4 \succ_D 3 \succ_D 2 \succ_D 1$$

**Algorithm 3** layer_follower

**Require:** $j, s_j, \mathcal{A}_p, \mathcal{R}_p, \mathcal{M}_p, M, p$
**Ensure:** $p, \mathcal{A}_p, \mathcal{M}_p, \mathcal{R}_p$
1: $\mathcal{A} \leftarrow \mathcal{A}_p, \mathcal{M} \leftarrow \mathcal{M}_p, \mathcal{R} \leftarrow \mathcal{R}_p$
2: **for** $m = 1, 2, ..., |\mathcal{M}_p|, i = 1, 2, ..., |\mathcal{A}_p|, i_1 = 1, 2, ..., M, i_2 = i_1 + 1, ..., M$ **do**
3:      **if** $s_j[\mathcal{A}_p[i]][i_2] == j$ **then**# agent $j$ is a receiver
4:          Agent $j$ pulls arm $i$, and sets $all\_leader\_success = 1 - C_j(t)$ if $s_j[\mathcal{A}_p[i]][i_1] \in \mathcal{M}_p$.
5:      **else**
6:          Agent $j$ pulls an arbitrary arm $i' \neq i$. # agent $j$ is a bystander
7:      **end if**
8: **end for**
9: **if** $all\_leader\_success == 1$ **then**
10:      Pull an arbitrary arm for $|\mathcal{M}_p|^2$ time steps.
11:      # wait for the layer leaders to look for their optimal stable matching arms
12:      **for** $\pi \in \mathcal{A}_p, \tau = 1, 2..., |\mathcal{R}_p| - |\mathcal{M}_p|$ **do**
13:          **if** $\tau == f$ ($f$ is the order of agent $j$ in layer follower set $\mathcal{R}_p \setminus \mathcal{M}_p$) **then**
14:              # it is agent $j$'s turn to explore available arms
15:              Agent $j$ pulls arm $\pi$
16:              **if** $C_j(t) == 1$ **then**
17:                  $\mathcal{A} \leftarrow \mathcal{A} \setminus \pi$
18:              **end if**
19:          **else**
20:              Agent $j$ pulls an arbitrary arm $\pi' \neq \pi$
21:          **end if**
22:      **end for**
23:      $\mathcal{R} \leftarrow \mathcal{R}_p \setminus \mathcal{M}_p, \mathcal{M} \leftarrow occ(j, \mathcal{A}, \mathcal{R}, s_j)$ #$\mathcal{M}$ denotes the set of new layer leaders
24:      $p \leftarrow p + 1$ #agent $j$ will later enter the next layer
25: **end if**
26: **return** $p, \mathcal{A}, \mathcal{M}, \mathcal{R}$

*From Definition 1 and 2, we can construct the communication graph $G(\mathcal{M}, \mathcal{A})$ and find out that the subset $\{A, B\}$ forms an outer closed circle. Thus, at layer $p = 1$, all agents will keep iterating between communication and exploration until the layer leaders $\mathcal{M}_1 = \{A, B\}$ achieve accurate estimations on all remaining arms $\mathcal{A}_1 = \mathcal{A} = \{1, 2, 3, 4\}$ and leave with their empirical optimal arms $\{1, 3\}$ (with high probability). Then, the layer followers $\{C, D\}$ will enter layer $p = 2$ with the arm set $\mathcal{A}_2 = \{2, 4\}$. With similar analysis, we conclude the whole process in Table 2. At the end (layer $p = 4$), all agents will exploit their optimal stable arms (with high probability).*

Table 2. Example 1

| $Layer(p)$ | $\mathcal{A}_p$ | $\mathcal{R}_p$ | $\mathcal{M}_p$ | $Exploitation$ |
|---|---|---|---|---|
| 1 | $\{1, 2, 3, 4\}$ | $\{A, B, C, D\}$ | $\{A, B\}$ | $\emptyset$ |
| 2 | $\{2, 4\}$ | $\{C, D\}$ | $\{D\}$ | $\{(A, 1), (B, 3)\}$ |
| 3 | $\{2\}$ | $\{C\}$ | $\{C\}$ | $\{(A, 1), (B, 3), (D, 4)\}$ |
| 4 | $\emptyset$ | $\emptyset$ | $\emptyset$ | $\{(A, 1), (B, 3), (C, 2), (D, 4)\}$ |

## 2.5 Regret Analysis

**Theorem 1.** *If every agent runs Algorithm ML-ETC, then the optimal regret after $T$ time steps is upper bounded by:*

$$Reg(T) \leq MK \lceil \frac{32}{\Delta^2} \rceil \lceil \log T \rceil + MC_1 + MC_2 + MC_3, \tag{1}$$

*where $C_1 = MK + M^4 K^2 \lceil \log M \rceil$, $C_2 = (\lceil \frac{32}{\Delta^2} \rceil + M)(\frac{KM^2(M-1)}{2} + M^2 + KM)$, and $C_3 = 2MK \lceil \frac{32}{\Delta^2} \rceil$.*

In Theorem 1, the regret term $MC_1$ is the communication cost induced by the initialization phase; the regret term $MC_2$ is the communication cost in the multi-layered exploration and communication phase; and the regret term $MC_3$ is the cost of wrong estimation under a low-probability event.

**Asymptotic term of** $T$: The majority term of regret is $MK\lceil\frac{32}{\Delta^2}\rceil\lceil\log T\rceil$, which is an $O(\log T)$ term and is better than all prior works [2, 10].

**Dependence of** $M, K$: The regret upper bounds of existing works (i.e., Phased-ETC and CA-UCB) have much higher dependence on $M, K$. For example, the regret bound of Phased-ETC contains a term of $O(\exp(\frac{1}{\Delta^2}))$. Note that $\Delta < 1/K$ (since the utility is from $[0, 1]$), this term is $O(\exp(K^2))$ and is much larger than ours. As for the regret bound of CA-UCB, it contains a term of $O(\exp(M^4)\log^2 T)$.

*Proof sketch:* The regret of our algorithm ML-ETC can be divided into three parts.

The regret caused by the initialization phase is upper bounded by $MC_1$, since the initialization phase lasts for $C_1$ time steps.

To bound the regret caused by the other two phases, we let $r_0 = r_1 = \lceil\frac{32}{\Delta^2}\rceil$, $r_2 = r_0 + M$, and define the following event.

$$\mathcal{E} = \left\{\forall j \in \mathcal{M}, r \leq r_0, m \in \mathcal{A}, |u_{jm}(t) - \hat{u}_{jm}(t)| \leq \sqrt{\frac{2\log T}{N_{jm}(t)}} \text{ holds after } r \text{ rounds of exploration}\right\}.$$

We prove that $\Pr[\mathcal{E}] \geq 1 - \frac{2r_0 MK}{T}$. Because of this, we know that the expected regret in the last two phases under $\neg\mathcal{E}$ is upper bounded by a constant $MC_3$.

Then we consider the expected regret in these two phases under $\mathcal{E}$. In this case, the length of the multi-layer exploration and communication phase can be upper bounded by $r_1 K\lceil\log T\rceil + C_2$, since the length of one round of exploration is $|A_p|\lceil\log T\rceil \leq K\lceil\log T\rceil$, the length of one round of communication can be upper bounded by $\frac{KM^2(M-1)}{2} + M^2 + KM$ and every agent will enter the exploitation phase after at most $r_1$ rounds of exploration and $r_2$ rounds of communication conditioning on $\mathcal{E}$.

On the other hand, under $\mathcal{E}$, all the agents will get the right estimation during the $GS\_and\_arm\_information$ procedure. Therefore, every agent $j$ must match with her optimal stable arm $a_j^*$ in the exploitation phase. This means that the expected regret in the exploitation phase under $\mathcal{E}$ is 0.

Summing over the expected regret in three phases, we finally get the regret upper bound in Theorem 1. Due to space limit, the details of the proof are deferred to Appendix. □

## 3 Simulation

**Setup**. We choose the time horizon to be $T = 2.5 \times 10^7$, arms' mean utilities within $[0.3, 0.6]$, and the minimal gap $\Delta = 0.05$. We have tested two cases with 5 agents and 5 arms but different preference and utility. To investigate the quality of the converging stable matching under different algorithms, we choose the arm preferences such that there exist multiple stable matches between agents and arms (see Appendix for the implementation detail).

**Baseline**. *Phased-ETC* [2] is a phased based decentralized algorithm which theoretically obtains the $O(\log^{1+\epsilon} T + \exp(\frac{1}{\Delta^2}))$ regret. The duration of each exploration is determined by the parameter $\epsilon$. When $\epsilon$ is smaller, the duration of each exploration is shorter. Same as the simulation in [2], we choose $\epsilon = 0.2$ in our simulation.

CA-UCB [10] is a UCB-based algorithm that aims to avoid collision combined. It guarantees an $O(\log^2 T)$ pessimal regret (a less strict notion of regret). Same as the simulation in [10], we choose the parameter $\lambda$ of delay probability to be $\lambda = 0.1$. Note that CA-UCB requires more information (i.e., all agents know the complete arm preference and can observe the winning agent for each arm at the end of each round), thus the comparison here is not entirely fair for our ML-ETC.

**Result**. Figure 1 shows the average regret and the standard deviation of regret over 50 independent runs. From Figure 1, ML-ETC outperforms both Phased-ETC and CA-UCB from an asymptotic view,

which shows that our proposed algorithm has lower regret in general cases, but may encounter a cold start. The cold start means that when the time horizon $T$ is small, ML-ETC may cause a relatively high regret. This is because ML-ETC has strict requirement of agents' estimation, only when the agents have very high confidence, will they enter the exploitation. As for Phased-ETC, the duration of the exploration is determined by the parameter $\epsilon$. The agents may start the exploitation only with limited exploration, and in this case they are not able to distinguish the arms, especially when $\Delta$ is small. Though this leads to a smaller short-term expected regret (at the beginning, Phased-ETC outperforms our ML-ETC), for large $T$, this can cause a very high expected regret of $O(\exp(\frac{1}{\Delta^2}))$ (in our experiments with $\Delta = 0.05$, this is even larger than $T$, and therefore the regret of Phased-ETC increases linearly as $T$ increases). In CA-UCB, there is possibility that an agent $j$ converges to pulling a sub-optimal arm $k \neq a_j^*$ in $\mathcal{A}_{stable}(j)$, which also causes a linear optimal regret.

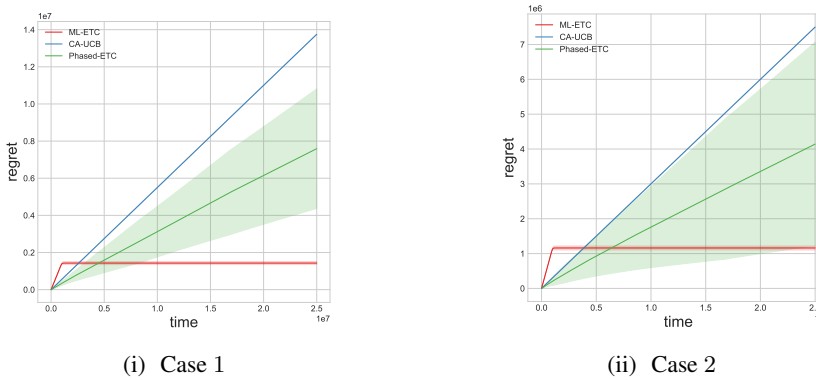

(i) Case 1          (ii) Case 2

Fig. 1. Comparison between ML-ETC and baselines.

## 4 Further Discussion on Possible Improvements

In this section, we discuss several possible improvements on ML-ETC, which may largely improve the efficiency of the communication and exploration.

**Initialization Phase**: In the $information\_access$ procedure, instead of conducting communication procedure $com(i, j, k, s_i)$ on all the "agent,agent,arm" tuples, we only need to conduct communication procedure $com(i, j, k, s_i)$ on some special tuples which satisfies that $j$ is ranked the next of $i$ on arm $k$. And this will reduce the regret of this procedure by $M$ times.

**Multi-layered exploration and communication Phase**: In the communication part for layer leaders to exchange signal of successful estimation, instead of conducting every effective communication procedure on every remaining arm, we can find a circle no longer than $2M$ that includes all layer leaders in the corresponding communication graph and conduct communication procedure through this circle. Specifically, we begin from an agent in this circle according to a pre-determined order and conduct effective communication along the circle until all agents have gone through the communication twice. This improvement will reduce the regret of this procedure by $O(M^2K)$ times.

**Dependence on the minimal gap** $\Delta$: In the exploration part, for example, instead of trying to distinguish all remaining arm, all layer leaders in layer $p$ only need to distinguish the $|\mathcal{M}_p|$-best remaining arms (which is the number of the layer leaders). This means that letting the agents only distinguish "meaningful arms" may also improve our algorithm by reducing the dependence on the minimal gap. We can use the minimal utility gap between "meaningful arms" instead of the minimal gap in our regret upper bound.

**The frequency of Communication**: Instead of exploring every remaining arms $\lceil \log T \rceil$ times per exploration, we can explore more time steps per exploration and reduce the frequency of communication, which may also help achieve lower regret.

## Acknowledgments and Disclosure of Funding

The work of Siwei Wang is supported in part by the National Natural Science Foundation of China Grant 62106122.

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
