# OpenReview forum: "Matching in Multi-arm Bandit with Collision"
_NeurIPS.cc/2022/Conference — NeurIPS 2022 Accept_

### Official Review · Reviewer_kPHe · 2022-06-28

**Rating:** 5
**Confidence:** 3
**Soundness:** 3 good
**Presentation:** 2 fair
**Contribution:** 2 fair

**Summary:**

The authors study the problem of bipartite matching under one-sided uncertainty with respect to preferences—i.e., where one set of market participants (here, agents) must learn their preferred ordering over the other set of market participants (here, arms) through repeated interactions (pulls). They do not assume that agents are globally ranked, but rather, that each arm has its own privately held preference order over agents, such that if multiple agents propose to the same arm at time $t$, a collision will occur, and only the agent most-preferred by the arm will obtain a non-zero stochastic reward. To address this setting, the authors introduce Multi-Layer Explore Then Commit (ML-ETC). This algorithm achieves asymptotic regret of $O(\log T)$, which is an improvement upon prior work in this space. Regret in this setting is defined with respect to the optimal stable matching (i.e., under Gale-Shapley with known preferences, rather than preferences derived vis-à-vis empirical means).

**Questions:**

1. This algorithm seems to require sustained adherence to the prescribed protocol(s) and trustworthy communication to be effective. Is it vulnerable to strategic behavior? (i.e., misrepresentation during the communication stage?)
2. Are arms’ preferences over agents required to be strict and complete?


**Limitations:**

- The authors acknowledge that one limitation of their proposed approach is that ML-ETC may outperform alternative approaches asymptotically, but over smaller, finite time horizons, the estimation accuracy requirements of ML-ETC may result in relatively high regret.
- A possible limitation is that the phase-based approach outlined here may result in uneven distribution of regret over agents. This concern could be mitigated if it’s possible to show that these agents would fare at least as poorly in expectation under existing approaches.


**Strengths And Weaknesses:**

*Strengths*:
- The problem considered— i.e., decentralized matching under uncertain preferences—is interesting, and the introduction of communication among (subsets of) agents in this setting seems reasonable/well-motivated.
- ML-ETC asymptotically outperforms existing SoTA approaches with respect to regret; empirical results support theoretical claims.
- Reproducibility: the authors provide detailed pseudocode along with source code.

*Weaknesses*:
- Scalability/ lack of sensitivity analysis: empirical results are reported only for small values of $M$ (number of agents) and $K$ (number of arms) and a single value of the minimal gap, $\Delta$.
- Clarity: while the paper is generally well-organized, certain details of the algorithmic exposition were a bit hard to follow (in particular, which communication(s) of what information were occurring simultaneously versus sequentially). A visual representation of the way in which arms and agents progress through the algorithmic pipeline over time/from phase to phase would be helpful. Additionally:
    - Since much of the algorithm builds upon Proposition 1, it would be helpful to include at least a proof sketch in the main paper (in particular, re: uniqueness).
    - (really minor) grammar suggestion: “switch from exploration to exploitation timely.” → “switch from exploration to exploitation in a timely fashion.”

---

> ### Author Response · Authors · 2022-08-02
> **-Reviewer kPHe  Thanks for your valuable comments! Here are our answers to your concerns.**
>
> -Reviewer kPHe
>
> Thanks for your valuable comments! Here are our answers to your concerns.
>
> Q1. Scalability / lack of sensitivity analysis.
>
> A1. We emphasize that the regret upper bounds of existing works (i.e., Phased-ETC and CA-UCB) have higher dependence on both $M$ and $T$ than ours.
> For example, the regret bound of Phased-ETC contains a term of $O(\exp({1\over \Delta^2}))$. Note that $\Delta \le {1\over M}$, this term is $O(\exp(M^2))$ and is much larger than ours $O(M^5)$. As for the regret bound of CA-UCB, it contains a term of $O(\exp(M^4)\log^2 T)$.
>
> We will clarify this in our revision, and include more experimental results to demonstrate the effectiveness of our ML-ETC algorithm.
>
> Q2. The details about communication protocol.
>
> A2. One communication procedure com(i,j,k,s) is occurring simultaneously.
>
> For example, consider a communication procedure com(A,B,1,(1,1,0)=6), i.e., agent A is transmitting message (1,1,0)=6 to agent B via arm 1.
>
> The whole communication will hold for 3 time steps. Agent A will pull arm 1 at time steps 1,2 after the communication begins, and pull another arm (not arm 1) at time step 3. Agent B will always pull arm 1, and other agents (i.e., the bystanders) will pull other arms (i.e., not arm 1). If arm 1 prefer A to B, agent B will encounter collision at time step 1,2, and encounter no collision at time step 3. Therefore agent B can get a binary string (1,1,0) according to the collision indicators, which is the same as the message that agent A wants to transmit.
>
> However, different com(i,j,k,s) are occurring sequentially. For example, in the 1 to $\log(M)$ time steps of the information_access in the initialization phase, com(A (agent 1),B (agent 2),1,1) happens.
>
> Q3. Proof sketch of Proposition 1.
>
> A3. Thanks for your advice, we will include a brief proof sketch in our revised manuscript.
>
> Q4. Does ML-ETC require sustained adherence to the prescribed protocol and trustworthy communication to be effective?
>
> A4. Since we mainly consider the collaborate agents rather than the competitive agents, we require the agents to follow the protocol. However, we don’t see obvious incentives for deviation since they will always converge to their optimal stable matching.
>
> As for the possible communication error (e.g., an agent mistakenly pulls an arm in practice), we can adopt other error correcting methods in communication to deal with them.
>
> Q5. Are arms' preferences over agents required to be strict and complete?
>
> A5. YES. Arms’ preferences over agents are required to be strict and complete.

---

> > ### Comment · Reviewer_kPHe · 2022-08-08
> > **Thanks for following up**
> >
> > Thank you to the authors for your detailed responses to my questions, as well as to those posed by the other reviewers. My primary outstanding concerns are related to scalability and clarity of presentation. I appreciate the regret bounds comparison to prior works, but do feel this theoretical claim should be supported by larger-scale empirical comparisons. I'm also curious about sensitivity analysis with respect to $T$ based on your conversation with Reviewer UKRH. For clarity of presentation, I do think a diagram or two would be quite helpful, and I'd also recommend pointing out in the paper or pseudocode which portion(s) of the algorithm can be parallelized, as well as indicating whether arms who have been matched are expected to continue supporting the communications/explorations of other arms (I may have misunderstood this aspect, but it was unclear to me when reading, and seemed like something that might require "compensation" to sustain collaboration).

---

> > > ### Author Response · Authors · 2022-08-09
> > > **Thanks for your help advice and comments**
> > >
> > > Thanks for your helpful advice and comments, we will improve our presentation and incorporate some diagrams for the process of  communication and matching. We will also add some disucssions on possible performance improvements/parallelization in the revision.

---

### Official Review · Reviewer_6TUF · 2022-07-03

**Rating:** 4
**Confidence:** 3
**Soundness:** 3 good
**Presentation:** 2 fair
**Contribution:** 2 fair

**Summary:**

The paper studies the problem of matching in multi-armed bandits with collisions, where $M$ agents interact with $K$ arms. Agents aim to maximize their (arm-dependent) reward, while arms have an ordinal preference over agents. If two agents play the same arm, the agent with the higher preference will gain the reward, while the second agent will experience a collision. As in some previous works, the authors use collisions to transmit information between agents. As the ‘winning’ agent does not observe a collision, some agents cannot receive messages from other agents, and the communication scheme can be described by a directional graph.

To solve this setting, the authors suggest a 3-phase algorithm. The first phase is a fixed-length initialization phase, in which all agents explore the graph structure and communicate it with each other. Then, agents are divided into a hierarchy that’s based on the graph structure. Agents on each level of the hierarchy explore all arms until they accurately identify them (‘exploration’), and then commit to a single arm until the end of the game. This is done sequentially at all levels of the agent hierarchy. The authors show that this algorithm (ML-ETC) achieves a logarithmic regret and empirically compare it to other baselines.

**Questions:**

Minor comments:
- Lines 37-38: when reading the stability definition for the first time, it seemed like the ‘both are more statisfied’ phrase referred to the two agents, and not the agent-arm. I suggest rephrasing it to be clearer.
- I would also appreciate it if the authors could say a bit more in the main text about the Gale-Shapely algorithm

**Limitations:**

The authors should discuss the weaknesses I described in greater detail (the synchronization assumption and implications of the dependence on the minimal gap).

**Strengths And Weaknesses:**

To my understanding, the main contributions of the paper are:
1) The characterization of the graph structure in the matching bandit problem,.
2) The insight that this structure effectively creates a hierarchy between agents that can be leveraged for layered exploration.

On the other hand, I think that there are a few major issues in this work. The main issue is the assumption that all agents are synchronized (always start the game at the same round). This assumption is not trivial in any multi-agent system, and to my knowledge, creates the `hack’ that allows the communication-through-collisions [4]. With this assumption, the agents can easily identify the graph structure (as done in the initialization stage) and make sure to avoid collisions except in designated communication phases. Therefore, synchronization makes the model much easier – there’s no major difference between a known or unknown preference model (and probably no real gain in knowing the identity of the agents we collide with).
A second weakness is the requirement to explore all arms for all agents until their ordering is identified with a high probability. This creates a dependence on the minimal gap between any two arms, which might be extremely small. In particular, the algorithm will probably break if any two arms have the same mean, and this also probably implies that the algorithm has no sublinear problem-independent regret.

Another smaller, related, issue is that the exploration phases must know the time horizon – the game is not only synchronized, but also cannot be anytime.


---------------------

### Post Rebuttal
I thank the reviewers for the response. I still don't think that the contribution (the communication characterization/protocol between agents) is sufficient to pass the acceptance bar, and I decided to leave my score unchanged.

---

> ### Author Response · Authors · 2022-08-02
> **-Reviewer 6TUF  Thanks for your valuable comments! Here are our answers to your concerns.**
>
> -Reviewer 6TUF
>
> Thanks for your valuable comments! Here are our answers to your concerns.
>
> Q1. About the synchronicity.
>
> A1. As far as we know, almost all the prior works that study matching in bandits adopt the assumption of synchronicity (e.g., [1][2]). The asynchronous case could be much more difficult and out of the range of our paper.
>
> Q2. Why all agents need to explore all the arms?
>
> A2. Thanks for pointing it out. We believe we can improve the algorithm a little by letting the agents only distinguish "meaningful arms" (for example, all the layer leaders only need to distinguish the $M_p$-best remaining arms, where $M_p$ is the number of the layer leaders). This means we can use the minimal utility gap between "meaningful arms" instead of the minimal gap $\Delta$ in our regret upper bound.
> Moreover, almost every explore-then-commit algorithm suffers from a dependency on such $\Delta$, e.g., [1].
>
> Q3. Our algorithm needs a known $T$.
>
> A3. If $T$ is unknown, then we can use the doubling trick.
>
> Q4. About the minor comments (writing and the Gale-Shapely algorithm).
>
> A4. We will improve the presentation, and explain more about stable matching and the Gale-Shapely algorithm in the final version. Thanks for your advice.
>
>
> References:
> [1] Basu, Soumya Sankar et al. “Beyond log2(T) Regret for Decentralized Bandits in Matching Markets.” ICML (2021).
>
> [2] Sankararaman A, Basu S, Sankararaman K A. Dominate or delete: Decentralized competing bandits in serial dictatorship[C]//International Conference on Artificial Intelligence and Statistics. PMLR, 2021: 1252-1260.

---

### Official Review · Reviewer_EhYJ · 2022-07-04

**Rating:** 5
**Confidence:** 4
**Soundness:** 4 excellent
**Presentation:** 3 good
**Contribution:** 2 fair

**Summary:**

This paper studies the matching under the multi-agent MAB setting. Specifically, the arms have preferences over agents; the communication is via a decentralized communication graph instead of via a central server. This paper proposes an ETC-based algorithm. The algorithm provably achieves state-of-the-art $\log(T)$ regret. Empirical study justifies the theory.

**Questions:**

The related work is not thoroughly discussed in this paper. To mention a few:

i. What is the relation between the problem considered in this paper with the classical combinatorial bandit problem (e.g. [1], [2])?

ii. Elimination-based algorithms are popular for MAB (e.g. [3]). Is it possible to leverage elimination-based methods in the setting of this paper and why or why not (intuition is fine)? My guess is that elimination probably does not help here due to the setting that arms have preference. But the authors should at least touch on this facet.

iii. MAB can be viewed as a special case of contextual linear bandit, and there are many related works studying two-sided matching with linear bandit (e.g. [4], [5], [6]). Should briefly mention how these works are related to this paper (e.g. similarity and difference in the setting and algorithm design; can the problem be reduced to each other)?

[1] Cesa-Bianchi, N. and Lugosi, G. Combinatorial bandits. Journal of Computer and System Sciences, 78(5):1404– 1422, 2012

[2] Chen, S., Lin, T., King, I., Lyu, M. R., and Chen, W. Combinatorial pure exploration of multi-armed bandits. In Ghahramani, Z., Welling, M., Cortes, C., Lawrence, N. D., and Weinberger, K. Q. (eds.), Advances in Neural Information Processing Systems 27, pp. 379–387. 2014.

[3] Katariya, S., Kveton, B., Szepesvari, C., Vernade, C., and Wen, Z. Stochastic Rank-1 Bandits. In Proceedings of the 20th International Conference on Artificial Intelligence and Statistics, volume 54 of Proceedings of Machine Learning Research, pp. 392–401, Fort Lauderdale, FL, USA, 20–22 Apr 2017.

[4] Learning Equilibria in Matching Markets from Bandit Feedback, M Jagadeesan, A Wei, Y Wang, M Jordan, J Steinhardt, 2021

[5] Learn to Match with No Regret: Reinforcement Learning in Markov Matching Markets, Y Min, T Wang, R Xu, Z Wang, M Jordan, Z Yang, 2022

[6] Rate-Optimal Contextual Online Matching Bandit, Y Li, C Wang, G Cheng, WW Sun

**Limitations:**

No extra concerns.

**Strengths And Weaknesses:**

Strengths:

1. The math is clear and easy for the readers to follow.

2. The paper contributes to a problem that is very important in ML theory and also in real applications.

Weaknesses:

1. The authors fail to discuss many important and probably highly related works.

2. The communication protocol is a bit complicated to understand. Also should justify why this protocol makes sense (by giving a simple example).

Overall, I think this paper makes good contributions to an important area. But the weaknesses make it hard to justify whether there are enough level of originality compared to the related work and thus the significance is not very clear at this moment. More elaboration is needed from the authors.

---

> ### Author Response · Authors · 2022-08-02
> **-Reviewer EhYJ  Thanks for your valuable comments! Here are our answers to your concerns.**
>
> -Reviewer EhYJ
>
> Thanks for your valuable comments! Here are our answers to your concerns.
>
> Q1. About the probably related works.
>
> A1. In fact these works are not similar to ours (please see Q3, Q4 and Q5), and therefore we do not mention them in our paper due to space limit. We appreciate you providing these related works. We will include them and discuss more related works in our final version.
>
> Q2. About the details of communication.
>
> A2. For example, consider a communication procedure com(A,B,1,(1,1,0)=6), i.e., agent A is transmitting message (1,1,0)=6 to agent B via arm 1.
>
> The whole communication will hold for 3 time steps. Agent A will pull arm 1 at time steps 1,2 after the communication begins, and pull another arm (not arm 1) at time step 3. Agent B will always pull arm 1, and other agents (i.e., the bystanders) will pull other arms (i.e., not arm 1). If arm 1 prefer A to B, agent B will encounter collision at time step 1,2, and encounter no collision at time step 3. Therefore, agent B can get a binary string (1,1,0) according to the collision indicators, which is the same as the message that agent A wants to transmit.
>
> Q3. What is the relation between the problem considered in this paper with the classical combinatorial bandit problem?
>
> A3. The settings in these papers are very different to our work. For example, in [1,2], the problem is a centralized problem, and the only agent knows about the global information in the system. Therefore, the main challenge is in the method of counting regret due to the combinatorial structure. However, our setting is a decentralized one, and each agent only knows about the partial information in the system. In this case, the main challenge is how can these agents collaborate with each other (e.g., by transmitting messages) to reduce the total regret.
>
> Q4. Would elimination-based methods (e.g., [3]) help in our setting?
>
> A4.  We believe that elimination-based algorithms may not be helpful in looking for the agent-optimal matching. The reason is that the matching arm of an agent $i$ depends on not only the arm utilities of agent $i$, but also the arm utilities of other agents $j\ne i$. Since the arm utilities of different agents are different, and one agent only knows about information about his utilities, it would be very complicated for him to eliminate some arm.
>
> Q5. About the similarities and difference between our model and two-sided matching with linear bandit.
>
> A5. As for similarities, both two settings consider the two-sides matching and need exploration to find good pairs of matching.
> Similar with combinatorial bandits [1,2], in the two-sided matching with linear bandit [4,5,6], there is a planner (platform) which takes actions for all participants and can observe the utilities of them, i.e., it is still a centralized setting. This is a major difference, since our setting is a decentralized setting.
>
>
>
>
> References:
> [1] Cesa-Bianchi, N. and Lugosi, G. Combinatorial bandits. Journal of Computer and System Sciences, 78(5):1404– 1422, 2012.
>
> [2] Chen, S., Lin, T., King, I., Lyu, M. R., and Chen, W. Combinatorial pure exploration of multi-armed bandits. In Ghahramani, Z., Welling, M., Cortes, C., Lawrence, N. D., and Weinberger, K. Q. (eds.), Advances in Neural Information Processing Systems 27, pp. 379–387. 2014.
>
> [3] Katariya, S., Kveton, B., Szepesvari, C., Vernade, C., and Wen, Z. Stochastic Rank-1 Bandits. In Proceedings of the 20th International Conference on Artificial Intelligence and Statistics, volume 54 of Proceedings of Machine Learning Research, pp. 392–401, Fort Lauderdale, FL, USA, 20–22 Apr 2017.
>
> [4] Learning Equilibria in Matching Markets from Bandit Feedback, M Jagadeesan, A Wei, Y Wang, M Jordan, J Steinhardt, 2021.
>
> [5] Learn to Match with No Regret: Reinforcement Learning in Markov Matching Markets, Y Min, T Wang, R Xu, Z Wang, M Jordan, Z Yang, 2022.
>
> [6] Rate-Optimal Contextual Online Matching Bandit, Y Li, C Wang, G Cheng, WW Sun.

---

### Official Review · Reviewer_UKRH · 2022-07-09

**Rating:** 3
**Confidence:** 3
**Soundness:** 1 poor
**Presentation:** 2 fair
**Contribution:** 2 fair

**Summary:**


This papers considers decentralized matching markets with bandit feedback. An algorithm, ML-ETC, is proposed that organises the agents in communicating/hierarchical layers. An upper bound on regret is proven and the algorithm is empirically compared against Phase-ETC and CA-UCB.

**Questions:**



- In line 30 you say that w.l.o.g. you assume that all gaps between the utility for all arms and agents is positive. Can you explain why this can be assumed "without loss of generality"?

- If I am not mistaken, your definition of regret is "incorrect" in that a lower bound would be linear. An arm can be attainable for agent $j$ according to your definition, however, under the agent-optimal matching $j$ may not actually be matched to $a^*_j$. Instead, you should probably say that an arm is attainable for agent $j$ if $j$ and this arm are being matched under the agent-optimal matching. (I could be wrong.)

- Line 98, you write that your algorithm achieves *asymptotic* regret $O(\log T)$. Please clarify.

- Line 283: You claim that your regret bound is $O(\log T)$ better than prior work. I don't think that this is true. See [1].



- As I already said, I have difficulties following the proof. Very weird is the step at line 510. First of all writing $\sqrt{\frac{\log T}{\log T}}$ makes no sense and is probably a typo(?). Secondly, what is the purpose (even when you add $N_{jn}(t)$) to the statement? Yes, you can control the distance of your estimated utilities to the true utilities in terms of the number of times that you saw these, however, I don't understand how you use this afterwards. The proof looks like the standard explore-first algorithm (which suffers larger regret than $\log T$). Lines 511 and following are very confusing to me.


- The fact that both Phase-ETC and CA-UCB suffer linear regret in your experiments is pretty surprising (even when you set the gap as small as you do). I couldn't find an implementation of these algorithms in the submitted code. Could you make it available? I'm curious about their empirical sensitivity to the gap on your problem instances.



- Minor: Line 171: $j_1 \in [1, M]$ should probably be $[M]$ ($= \{1, \dots, M\}$)



[1] *Beyond log2 (T) Regret for Decentralized Bandits in Matching Markets*, Basu et al.

**Limitations:**

.

**Strengths And Weaknesses:**


The authors claim a regret bound of order $O(MK \log(T) / \Delta^2 + M^5 K^2)$, which would, as far as I can tell, be an improvement over prior work (Although $M^5 K^2$ is a fairly large overhead compared to e.g.\ the dependence on $M$ and $K$ in [1].)

However, I have concerns about the soundness of the paper, both on the theoretical as well as the empirical side.
To be honest, I cannot really make sense of the proof given in the Appendix and I'd like to hear my fellow reviewers' opinion on the proof of Theorem~1. Moreover, I'd like to ask the authors to elaborate on the empirical results (see below).


The organisation of the paper is fine, although the writing would benefit from a few more iterations.

I am willing to increase my score if the soundness of the paper can be demonstrated by means of a rigorous, easier to follow proof and empirical reproducibility.

[1] *Beyond log2 (T) Regret for Decentralized Bandits in Matching Markets*, Basu et al.

---

> ### Author Response · Authors · 2022-08-02
> **Reviewer UKRH: Thanks for your valuable comments. We’ve addressed the typo and answered the questions about the proof below, and we are happy to answer any further questions.**
>
> - Reviewer UKRH:
>
> Thanks for your valuable comments. We’ve addressed the typo and answered the questions about the proof below, and we are happy to answer any further questions.
>
> Q1. (In line 30) Why $\Delta>0$ is assumed "without loss of generality"?
>
> A1. We’re sorry for the confusing presentation, it is indeed a commonly adopted assumption (e.g., [1],[2]). We will clarify this in our final version.
>
> Q2. About the definition of regret.
>
> A2. We emphasize that our definition of regret is *correct*. From the property of stable matching, under the agent-optimal matching, agent $j$ will be matched with $a_j^*$. ([4]). This definition is also used in many prior works, e.g., [1][3].
>
> Q3. (In line 98) Our algorithm achieves *asymptotic* regret $O(\log T)$.
>
> A3. This is a typo. It should be "achieves $O(\log T )$ regret". We will fix it in our final version.
>
> Q4. (In line 283) We claim that our regret bound is $O(\log T)$ better than prior work.
>
> A4. We think this is a misunderstanding. In line 283, by the sentence "which is an $O(\log T)$ term and better than all prior works [2, 9].", we only claim that our regret bound is better than prior works *but not* "$O(\log T)$ better".
>
> Q5. About the steps at line 510.
>
> A5. We are sorry that there are some typos and we omit some steps in these lines. This may make the steps a little confusing, and lead to your misunderstanding.
>
> The line 509-511 should be:
>
> Condition on $\mathcal{E}$, after $r_0$ rounds of explorations, for any $j\in \mathcal{M}$, $m,n \in \mathcal{A}\_p$ such that $u_{jm} > u_{jn}$, we have
>
> $\hat{u}\_{jm} - \sqrt{2\log T \over N_{jm}(T)} \ge u_{jm} - 2\sqrt{2\log T \over N_{jm}(T)} \ge u_{jn} +\Delta- 2\sqrt{2\log T \over N_{jm}(T)} \ge u_{jn} +2\sqrt{2\log T \over N_{jn}(T)} + \Delta- 2\sqrt{2\log T \over N_{jm}(T)} - 2\sqrt{2\log T \over N_{jn}(T)}$
> $ \ge \hat{u}\_{jn}  +\sqrt{2\log T \over N_{jn}(T)} + \Delta- 2\sqrt{2\log T \over N_{jm}(T)} - 2\sqrt{2\log T \over N_{jn}(T)} = \hat{u}\_{jn}  +\sqrt{2\log T \over N_{jn}(T)} + \Delta- 4\sqrt{2\log T \over r_0\log T}$
>
> Here the last equation is because that after $r_0$ rounds of explorations, all the remaining arms have $r_0\log T$ observations, and therefore $N_{jm}(T) = N_{jn}(T) = r_0\log T$.
>
> This inequality implies that after $r_0$ rounds of exploration, all the agents can distinguish the remaining arms with high confidence, and therefore they will send the message that their exploration succeeds after at most $r_0$ rounds of exploration.
>
> Q6. Why do both Phase-ETC and CA-UCB suffer linear regret in our experiments?
>
> A6. The reason why CA-UCB suffers linear regret is that it may not converge to the agent-optimal matching. For example, in the 3-agent and 3-arm simulation in our work (line 535 of the supplementary file), CA-UCB will converge to the matching ${(A,2),(B,3),(C,1)}$ rather than the optimal stable matching ${(A,3),(B,1),(C,2)}$.
> As for Phased-ETC, since the number of explorations only depends on $T$ but not $\Delta$, when $\Delta$ is small, there is a high probability for Phased-ETC to estimate wrong and not converge to agent-optimal matching. This also results in a linear regret.
>
> References:
> [1] Basu, Soumya Sankar et al. “Beyond log2(T) Regret for Decentralized Bandits in Matching Markets.” ICML (2021).
>
> [2] Liu L T, Ruan F, Mania H, et al. Bandit learning in decentralized matching markets[J]. Journal of Machine Learning Research, 2021, 22(211): 1-34.
>
> [3] Liu L T, Mania H, Jordan M. Competing bandits in matching markets[C]//International Conference on Artificial Intelligence and Statistics. PMLR, 2020: 1618-1628.
>
> [4] Kleinberg, Jon, and Eva Tardos. Algorithm design. Pearson Education India, 2006.

---

> > ### Comment · Reviewer_UKRH · 2022-08-05
> > **Response to Rebuttal: some unanswered questions**
> >
> > Thank you very much for your response. Some things (in particular those due to typos were clarified). However, I still have some comments and some of my questions remain unanswered:
> >
> > 1. Definition of regret. I think that you do not define regret the same way as [1] and [3]. You probably intended to, but what you wrote is not the same. Suppose that there exists a stable matching where agent $i$ is matched with arm $k$ and another stable matching where agent $j$ is matched with arm $k$. Suppose that $a_i* = k$ as well as $a_j^* = k$ (in your notation) and further let $u_{i,k}, u_{j,k} > u_{x, y}$ for all other $x, y$. Then, the first sum in your definition of regret includes both $u_{i,k}$ and $u_{j,k}$. However, obtaining both is impossible under any single matching. In other words, the first term in your regret (the way you define it) is not restricted to matchings only.
> >
> > 2. Why would CA-UCB converge to a suboptimal matching?
> > If the algorithm would really suffer linear regret, what about the sublinear regret bounds they prove?
> > *Moreover, you did not respond to my request for the actual code that was used for the experiments. I have serious doubts that both CA-UCB and Phase-ETC would not improve at all over time. To underline the soundness of your work, please make the code you used to create the figures available.*

---

> > > ### Author Response · Authors · 2022-08-08
> > > **Thank you very much for your response.**
> > >
> > > 1. First, we recall the definition of attainable arms, the optimal-stable arm and the pessimal-stable arm. If an arm $k$ is an attainable arm for agent $i$, it means that there exists at least one stable matching where arm $k$ is matched with agent $i$. The optimal-stable arm $a_i^*$ for agent $i$ is the most preferred attainable arm for agent $i$, which means $a_i^*$ is the one with highest utility among all these attainable arms. The pessimal-stable arm $\underline{a}_i$ for agent $i$ is the least preferred attainable arm for agent $i$, which means $\underline{a}_i$ is the one with lowest utility among all these attainable arms.
> > > Second, as the key property of stable matching, [5] have shown that for any given both sides’ preferences, the agent-optimal stable matching exists and is unique. More specifically, each agent’s most preferred attainable arm will be different, and every agent will be matched with their most preferred attainable arm in the agent-optimal stable matching. Thus, there does not exist agent $i \ne j$ such that $a_j^*=a_i^*$ (where $a_i^*$ is the most preferred attainable arm for agent i).
> > > Third, our definition of regret is not exactly the same with the definition of optimal regret in [1],[3]. However, the only difference is that we sum up the regret of all agents together.
> > >
> > > 2. The regret considered in [2] is a less strict definition of regret, i.e., the pessimal regret, which is different from the optimal regret considered in our paper. The pessimal regret is the expected utility gap between the pessimal-stable arm (instead of the optimal-stable arm) and the achieved matching, which means:
> > > $$\underline{Reg(T)}= T\sum{u\_{i \underline{a}_i}}-E[\sum {(1-C_i(t))u\_{iI_i(t)}}]$$
> > > As for CA-UCB([2]), they proved the bound of pessimal regret instead of the (optimal) regret we use in our work. Due to the definition of the pessimal regret and the (optimal) regret we use, we can easily conduct that the (optimal) regret we use is always no bigger than the pessimal regret and there will be a linear gap between them when the optimal-stable arm does not equal to the pessimal-stable arm for an agent. So there is no contradiction between their work and our simulation.
> > > As for Phased-ETC([3]), with small T, its exploration is not enough to distinguish the arms, this will cause false exploitation and cause nearly linear regret (the $exp(\Delta)$ factor in their bound shows this situation).
> > > We have now uploaded our code for the benchmark algorithms in the revision.
> > >
> > > References:
> > > [1] Basu, Soumya Sankar et al. “Beyond log2(T) Regret for Decentralized Bandits in Matching Markets.” ICML (2021).
> > >
> > > [2] Liu L T, Ruan F, Mania H, et al. Bandit learning in decentralized matching markets[J]. Journal of Machine Learning Research, 2021, 22(211): 1-34.
> > >
> > > [3] Liu L T, Mania H, Jordan M. Competing bandits in matching markets[C]//International Conference on Artificial Intelligence and Statistics. PMLR, 2020: 1618-1628.
> > >
> > > [4]Kleinberg, Jon, and Eva Tardos. Algorithm design. Pearson Education India, 2006.
> > >
> > > [5]Gale D, Shapley L S. College admissions and the stability of marriage[J]. The American Mathematical Monthly, 1962, 69(1): 9-15.

---

### Official Review · Reviewer_gFbZ · 2022-07-12

**Rating:** 6
**Confidence:** 2
**Soundness:** 3 good
**Presentation:** 3 good
**Contribution:** 3 good

**Summary:**

The paper is concerned with matching and MAB setting of Liu et al, 2020. The main contribution of this paper is an improvement in the regret to O(log(T)) through a communication protocol between the agents.

**Questions:**

In particular, I would the authors to address points 1 and 2 below weaknesses in the above section.

Minor Issue: In Line 24, I_j(t) \in A (set of arms), not M (set of agents).

**Limitations:**

Authors have adequately the limitations

**Strengths And Weaknesses:**

$\textbf{Strengths}:$

-This setting has gained significant attention and I think an improvement to log(T) is significant. Further the algorithm experimentally outperforms the other baselines.

-It does not seem that using communication between agents was explored in this setting.

$\textbf{Weaknesses}:$

1-The regret seems to have a higher linear dependence on the number of agents M. Is this a drawback? For example the simulation results actually involve a small number of agents (3 and 5). Would the algorithm still outperform other baselines when the number of agents is increased, it seems that it would be good to demonstrate that.

2-While the agent only needs knowledge of his history and collisions to make decisions, is the setting completely decentralized? Is it obvious that no agent has an incentive to deviate from the protocol?

3-The discussion of the algorithm discussion is rather  verbose. While this is not necessarily a weakness, I think the technical content could be improved representation-wise perhaps through the addition of diagrams.

---

> ### Author Response · Authors · 2022-08-02
> **- Reviewer gFbZ: Thanks for your valuable comments! Here are our answers to your concerns.**
>
> - Reviewer gFbZ:
> Thanks for your valuable comments! Here are our answers to your concerns.
>
> Q1. The regret seems to have a higher linear dependence on the number of agents $M$.
>
> A1. In fact this is not a drawback, since the regret upper bounds of existing works (i.e., Phased-ETC and CA-UCB)  have much higher dependence. For example, the regret bound of Phased-ETC contains a term of $O(\exp({1\over \Delta^2}))$. Note that $\Delta \le {1\over M}$, this term is $O(\exp(M^2))$ and is much larger than ours. As for the regret bound of CA-UCB, it contains a term of $O(\exp(M^4)\log^2 T)$.
>
> The above explanations show that our algorithm is better in both the dependence on $T$ and the dependence on $M$. Therefore our algorithm will outperform other baselines for large $T$ or large $M$. We will clarify this in our revision, and include more experimental results to demonstrate the effectiveness of our ML-ETC algorithm.
>
> Q2. Is the setting completely decentralized? Is it obvious that no agent has an incentive to deviate from the protocol?
>
> A2. i) The setting is completely decentralized.
> ii) Since every agent will converge to their optimal stable matching, we don’t see obvious reasons for agents to deviate. Whether agents have the incentive to deviate the protocol is an interesting issue, but it is out of the research topic of our paper, since we mainly consider collaborative agents rather than competitive agents here. We may leave it as one of our future works.
>
> Q3. About the writings.
>
> A3. We will improve the presentation, and fix typos in our revision.

---

> > ### Comment · Reviewer_gFbZ · 2022-08-08
> > **Follow up**
> >
> > Thank you for your response. I was not aware of the large dependence on the number of agents in previous work, I think the paper would benefit from pointing this out.

---

> > > ### Author Response · Authors · 2022-08-09
> > > **Thanks for your helpful advice and comments**
> > >
> > > Thanks for your helpful advice and comments, we will include the comparison of the dependence on the number of agents in our final version.

---

### Meta-Review · Area_Chair_SEa8 · 2022-09-01

**Recommendation:** Accept
**Confidence:** Certain

**Metareview:**

This paper is concerned with multi-player multi-armed bandit with arm preferences. It improves the existing literature by improving the regret upper-bound from log^{1+eps}(T) to log(T).

The main ingredient is, as usual in the literature, forced collision. The issue in the model is that since arms have preferences, some communication between players might be impossible (if all arms always prefer a single player to another one). The trick is then to add a phase to "discover" what kind of communication is possible (via collision) and then this issue can somehow be circumvented.

This idea is quite elegant, and the paper quite clear (even though the lack of space makes it quite difficult for the reviewers to understand the protocols involved).

The major concern was the incrementality of this paper with respect to the existing literature: the trick is nice, but maybe not breathtaking.

I hesitated quite a lot, discussed with colleagues, and finally decided that competing/matching bandits is a nice and intriguing setting that deserve to be investigated more. Hence I recommend acceptance.

**Award:**

No

---

### Decision · Program_Chairs · 2022-09-14

Accept